# Caregivers' experience of seeking care for adolescents with sickle cell disease in a tertiary care hospital in Bahrain

**Khadija Al Saif**[1], **Fatema Mohamed Abdulla**[1], **Anwaar Alrahim**[1], **Sara Abduljawad**[1], **Zainab Matrook**[1], **Jenan Jaafar Abdulla**[1], **Fatima Bughamar**[1], **Fatema Alasfoor**[1], **Rana Taqi**[1], **Amna Almarzooq**[1], **Jamil Ahmed**[2]*

1 College of Medicine and Medical Sciences, Arabian Gulf University, Manama, Kingdom of Bahrain,
2 Department of Family and Community Medicine, College of Medicine and Medical Sciences, Arabian Gulf University, Manama, Kingdom of Bahrain

* jamilmga@agu.edu.bh, jamil.ahmed.dr@gmail.com

## Abstract

### Objective

This study aimed to determine caregivers' perspectives on difficulties encountered while seeking care for adolescents with sickle cell disease (SCD). It explored the social, emotional, and financial impact of caring for an adolescent with SCD on their caregivers.

### Study design

A mixed-method study in a major tertiary care hospital in Bahrain was conducted between June and August 2019. Cross-sectional questionnaires and thematic analyzed interviews were performed with 101 and 18 Bahraini caregivers of adolescents with SCD (aged 10–18 years), respectively.

### Results

Lack of parking lots (52.5%) and traffic jams (27%) were identified as the most common challenges in seeking hospital care for adolescents with SCD. These difficulties, including prolonged waiting in the emergency room, discouraged more than half of the caregivers who preferred to seek care from smaller healthcare centers. Most caregivers reported receiving a high degree of support from their families, who emotionally encouraged them to facilitate patient care (73.3%). Therefore, their relationships with their friends, colleagues, and relatives were not significantly affected. Catastrophic health expenditure occurred in 14.8% of caregivers. Qualitative themes that emerged were A) *the intricacy of caring for adolescents with SCD*, B) *dissatisfaction with hospital facilities*, and C) *insufficient healthcare services*, wherein caregivers reported adolescents' experiences with services during hospital visits. Subthemes for the intricacy of caring for adolescents with SCD were 1) *the psychological tragedy*, summarizing the initial caregivers' feelings after the confirmed diagnosis, 2) *caregiving hardships* that described the caregivers' emotional and health burden

**Data Availability Statement:** All relevant data are within the paper and its Supporting Information files.

**Funding:** The authors received no specific funding for this work.

**Competing interests:** The authors have declared that no competing interests exist.

while accepting and adjusting to the disease, and 3) *the cost of care on families*, which highlights the financial burden of the disease on families.

## Conclusion

The caregivers of adolescents with SCD experienced several overwhelming challenges, including problems in accessing healthcare and receiving medical services, in addition to influences on the emotional, financial, and social aspects of their lives.

## Introduction

Sickle cell disease (SCD) is a group of inherited disorders of red blood cells associated with several health conditions and typically begins at 5 to 6 months of life [1]. About 300,000 to 400,000 infants are affected by SCD every year, mostly in Sub-Saharan Africa, where the incidence of newborns with SCD in 2010 was about 230,000 [1]. It is an autosomal codominance trait that runs in families and is also relatively more prevalent in the Middle Eastern region due to high rates of consanguineous marriages [1–3]. The prevalence of newborns with SCD in Bahrain has been reported to be approximately 2.1% in 1985 and 0.4% in 2010 [4, 5]. In 2021, the prevalence was 0.2% [6]. SCD is one of the most common reasons for hospital admissions in Bahrain and has the highest rate for multiple readmissions [7]. According to the ministry of health statistics report in 2017, SCD was the most common cause of admissions in the main tertiary hospital in the country and was reported to be 10.5% of admissions [8]. The mortality rate for Bahraini patients with SCD in intensive care can be as high as 12.7% [7].

Patients with SCD frequently experience attacks of pain called "Sickle Cell Crisis," anemia, acute chest syndrome, swelling in the hands and feet, bacterial infections, and even stroke [1]. Over a patient's lifetime, SCD can cause organ damage involving the spleen, brain, eyes, lungs, liver, heart, kidneys, penis, joints, bones, and skin [9–12]. As children with SCD move into adolescence, sickle cell pain becomes recurrent, acute, and unpredictable [13]. The sickle cell crisis, a sudden appearance of body pain lasting for several hours resulting from vaso-occlusive complications, is the main challenge for both patients and their caregivers, leading to poor psychological well-being and overall health of patients and their caregivers [14–16].

A study has noted that caregivers experience shock after the diagnosis of their child [17]. Studies have also reported a higher incidence of depression, accompanied by significant guilt and fear of having another child with the same disease, among parents of children with SCD [18]. Caring for an adolescent with a chronic illness adds significant stress and responsibility to caregivers, affecting them emotionally and mentally [19]. Anxiety, depression, exhaustion, and poor sleep are the most common ailments reported by caregivers [20]. Apart from the heavy emotional and mental toll, caregivers' social lives are also impacted [21]. In addition, caregivers spend less time with unaffected siblings [21, 22]

Caring for a person with SCD can impose a large financial burden on the caregivers. Catastrophic health expenditure refers to a situation in which a household's health spending surpasses basic needs and/or causes debts, sells assets, and becomes poor [23–28]. The widely used threshold to define catastrophic health expenditure is if spending on health is more than 10% of total household income or more than 40% of household non-food expenditure [23, 27, 29, 30]. Catastrophic health expenditures can account for as much as 20.7% of household spending among patients with SCD, using 10% of total income as the threshold level. The chances of reaching catastrophic health expenditure are higher among parents with poor

socioeconomic status who take loans to pay hospital fees. In addition, about 2.7% of parents lose their jobs to take care of their affected children and are unable to commit to work. In addition, the indirect cost of missing working days could be large [23]. As many as 63.1% of caregivers experience financial difficulties [21].

Access to healthcare services can be a challenge for patients with SCD and their families, leading to poor a quality of life. Studies have reported that families of children with SCD prefer visiting the local hospital because of the great distance from a specialized sickle cell center [31]. Caregivers complain of a lack of clinical knowledge and expertise in treating patients with SCD among local hospital physicians [22, 31]. Currently, Bahrain has a hereditary blood disorders center that solely admits males [32]. Other complaints include long waiting times before receiving care in the emergency department [32, 33]. Therefore, patients with SCD commonly prefer to treat the symptoms of SCD outside of the healthcare setting [33]. It is also noted that healthcare workers stigmatize patients as drug addicts, and complaints by patients with SCD may not be taken seriously [22, 33].

Studies have sparsely focused on how caring for children and adolescents with chronic disorders affects caregivers. In the Middle Eastern region, including Bahrain, where the burden of genetic and inherited disorders requiring long-term care is high, there is limited evidence on this topic. Therefore, evaluation of the impact of being a caregiver for an adolescent with SCD is considered a valuable research area. Adolescence is a phase of life between childhood and adulthood, from ages 10–19 [34]. The adolescent age group was targeted because adolescence is a sensitive and critical period of change in the patients' lives where they experience rapid physical, psychological, social, and behavioral changes and require considerable parental guidance and support [35]. Further, adolescents are more prone to adverse symptoms of SCD [36]. Caregivers play an essential role in supporting adolescents' adjustment to the disease [37]. Although studies conducted elsewhere were informative, the experience of caregivers can differ since other healthcare systems have different facilities and may provide various types and levels of services. This study aimed to understand caregivers' perceptions of their interactions with the healthcare system and to map their barriers to healthcare accessibility. In addition, it aimed to explore caregivers' experiences of having an adolescent with SCD and its impact on the social, emotional, financial, and health aspects of caregivers' lives. It is expected that this study will empower caregivers to discuss their experiences caring for adolescents with SCD. This study could be used as a resource to improve the healthcare system and the well-being of individuals with SCD and their families. These include better awareness of their difficulties, more understanding of the situation, and helpful insights on how to provide support and appropriate services to them and their caregivers.

## Methods

A mixed-method study design was employed. The qualitative research method was included to complement the quantitative design for an in-depth understanding of the research topic.

### Study setting

The study was conducted at the Salmaniya Medical Complex (SMC), the largest tertiary care hospital in Bahrain, from June to August 2019. It is a major public hospital that provides comprehensive medical services, including emergency, secondary, tertiary, and outpatient healthcare services. SMC has a capacity of approximately 1200 beds and receives an average of 900–1000 patients per day (20). Healthcare services in public hospitals in Bahrain are provided free of charge to all Bahrainis and Gulf Cooperation Council residents. A premarital screening program was introduced in Bahrain in 1985 and expanded in 1992 to reduce the incidence of

genetic diseases. By law in 2004, Bahraini couples intending to marry must undergo compulsory premarital screening. Later, premarital counseling was included as a part of primary health care services in all healthcare centers throughout Bahrain. In addition, a newborn screening program for the detection of blood disorders was initiated in 2007 [38].

### Selection of participants

The study participants were primary caregivers of adolescents aged 10–18 years with SCD who were admitted at SMC during the study period. A caregiver was defined as the person who cared for an adolescent with SCD and took most of the responsibility and guardianship regarding the adolescent's treatment during an SCD crisis. Participants were identified and recruited with the help of nurse supervisors of the wards and outpatient clinics from a list of registered patients using a non-probability consecutive sampling technique. In addition to the previous sampling technique, some participants in quantitative study were added by snowball sampling, wherein caregivers were asked if they knew other caregivers who met the inclusion criteria. Participants who did not speak Arabic were excluded to avoid language barriers. Caregivers were also excluded if the Patient with SCD was less than 10 years or older than 18 years. Participants were selected with a ratio of one caregiver to one SCD patient.

### Study design and sample

Newborns from 2001–2009 were aged 10–18 at the time of this study. A total of 181 newborns were born with SCD in Bahrain from 2007 to 2009 [5]. The National Society for Hereditary Blood Disease screening program mentioned that among all Bahrain public and private schools, there were between 60–70 students in each academic year in 2014 [39]. The population size was derived from the average number of SCD students each academic year, which was 65 for newborns in 2001–2006, and with the available statistics for newborns from 2007–2009; the population size was thus estimated to be n = 571.

Based on power estimation, the sample size was calculated using the following formula [40]:

$$n_0 = \frac{Z_{\alpha/2}^2 \times P \times (1 - P)}{E^2}$$

$$n = \frac{n_0}{1 + \frac{n_0 - 1}{N}}$$

E = Margin of error (9%)
Z = Critical value with 5% level of significance (1.96)
P = Population portion (0.5)
N = estimated population (571)
$n_0$ = infinite population (119)
n = sample size (99)

A cross-sectional survey was used to understand the challenges faced by caregivers of Bahraini adolescents with SCD in accessing healthcare, and to understand how caring for an adolescent with SCD impacted caregivers' lives. The complementary qualitative approach was expected to allow caregivers to reveal their experiences in dealing with difficulties and express their views regarding how their children's conditions affected their own lives. Most caregivers were interviewed in the outpatient clinics. A total of 101 caregivers participated in the quantitative analysis. In total, 62 and 38 caregivers caring for an adolescent with SCD in early (10–14 years) and late adolescence (15–18 years), respectively, answered the questionnaire. As for the qualitative analysis, 18 caregivers—15 from medical wards and 3 from outpatient clinics—

were interviewed; 11 cared for a patient with SCD in early adolescence (10–14 years), and 7 cared for a patient in late adolescence (15–18 years). Of the 101 caregivers in the quantitative study, only two were included in the qualitative study.

## Data collection

Data were collected through interviews conducted by ten Bahraini fourth-year medical students from the Arabian Gulf University who received training on data collection from the senior author. Before interviewing, the purpose of the investigation was explained to the participants. Next, interviews were conducted in a room provided by the hospital to ensure the confidentiality of the data collection.

## Quantitative data collection

The researchers AnA, JJA, FB, FA, and AmA interviewed and filled the questionnaires in Arabic. Each interviewer completed 20 questionnaires, which took approximately 10 to 20 minutes to answer then checked the questionnaires for incomplete data. Questionnaires that did not fulfill the purpose and objectives of the study were not collected. Questions that caregivers could not fill out because the answers differed at various times and questions that were sensitive to the caregiver were omitted from calculations, but these caregivers were included in the study.

A structured questionnaire was developed according to the objectives of our study by reviewing the literature. During questionnaire development, researchers created a draft questionnaire and then performed brief qualitative interviews with caregivers to modify the questions. The questionnaire was reviewed by the senior author to ensure that the content was appropriate and relevant. A pilot study was conducted to test the validity of the questionnaire. The pretested sample of ten participants suggested no difficulties in comprehending or answering the questions. The Cronbach's alpha of this set of questions was 0.771.

The data collection tool included items on the socio-demographic characteristics of caregivers and patients and evaluation of healthcare access during a sickle cell crisis. We asked questions such as "What is your first destination when the patient has a sickle cell crisis," "Why did you choose it as a first destination," and "What are the challenges you face in your way to the hospital." To assess the influence of being a caregiver on their social, emotional, and financial life, questions such as the following were asked, "On a scale of 1 to 5, how do you personally evaluate the support you receive from your family," "How does the condition of your child affect your relationship with each of the following people: partner, family member, other children, friends, co-workers" and "On average how much do you spend on the patient's treatment in a month?" The degree of support was measured on a scale of one to five and was recorded as low (1–2), intermediate (3), or sufficient (4–5). Caregivers' relationships were divided into five categories: relationships with partners, family members, other children, friends, and coworkers. The impact of caregivers' relationships was considered either low or high if less than or more than two categories, respectively, were affected. The final general assessment section was the calculation of the "catastrophic health expenditure": Percentage income spent as health expenditure($T_{HE/FI}$) = ($HE_T/T_{FI}$) × 100, where $HE_T$ and $T_{FI}$ represent the total health expenditure and total household (family) income, respectively, during the month [23]. Any household with a health expenditure proportion ($T_{HE/FI}$) ≥10% was designated as having suffered catastrophic health expenditure.

## Qualitative data collection

The researchers KA, FMA, SA, ZM, and RT conducted interactive in-depth interviews using a semi-structured in-depth interview guide in Arabic. Each interviewer completed two

interviews, lasting approximately 40 minutes each. Caregivers were informed that discussions would be recorded and could be stopped if discomfort occurred.

The qualitative interviews included open-ended questions such as, "Can you tell me about the difficulties you face while trying to seek healthcare for your child," "How do you feel about taking care of your child/adolescent with SCD," "How does it affect your relationship with your family and friends," and "How did your child's condition affect your financial status," and probes such as "Can you tell me more about that?" By the time 18 primary caregivers were interviewed, data saturation had been reached, and the study team agreed to discontinue any further interviews.

## Data analysis

For the quantitative information the researchers AnA, JJA, FB, FA, and AmA translated and entered data into Microsoft Excel, and then assigned investigators double-checked these for accuracy. Later, data were imported into the Statistical Package for the Social Sciences version 20, where they were coded and analyzed. For the descriptive statistics, quantitative variables, such as mean and standard deviation, were calculated to describe the age of caregivers and patients. For other variables, proportions or absolute values were presented.

For the qualitative information, data were analyzed through thematic content analysis. Audio-recorded interviews were transcribed verbatim by the researchers KA, FMA, SA, ZM, and RT. Manual coding of the Arabic interviews was performed in a table format, and codes were applied to the text in an adjacent column. After a consensus was reached between the study team, the codes were revised and finalized. The generation of codes depended mainly on the importance of the data and/or repetition of the same data in several interviews. During the analysis, codes were translated into English and reviewed sequentially with a qualitative research expert for better accuracy and rationality of patterns. Codes led to the generation of categories and themes. The representative quotes under the codes were translated from Arabic to English.

## Ethical consideration

The study was approved by the Research and Ethics Committee of the Arabian Gulf University (No E008-PI-4/19) and the Health Research Committee of the Ministry of Health in Bahrain (No AURS/276/2019). Administrative permission was obtained from SMC. Written informed consent was obtained from the caregivers. Caregivers were informed that they had the freedom to withdraw at any time during the study, and their agreement or refusal would not interfere with the healthcare of the adolescent in the hospital. Names were replaced with their initials, and other personal information about caregivers was concealed to maintain privacy. Participants were assured that they had adequate time during the interview to ease stress, reveal their fears, and eliminate hesitation while considering their body language and facial expressions. The researchers expressed empathy and gave participants complete attention during the interviews.

## Results

### Quantitative results

Six questionnaires with significant missing data were excluded. Only data from complete questionnaires were analyzed. Approximately 67% of the caregivers were Bahrainis aged 40 to 50 years with a mean age of 44.24 years (SD± 6.25); 77.2% were female, 92.1% were married, and 73.3% were mothers of the patients. About 48.5% of the caregivers were unemployed, and

41.6% were high school graduates. Among the Patients with SCD, 41.6% were 13 to 15 years old, while 32.7% of them were 10 to 12 years old; the mean age of participants was 13.73 years (SD ± 2.37) (Table 1).

About half (52.5%) of the caregivers complained about the lack of parking space at the hospital, while 26.7% stated that they experienced traffic jam problems on their way to the hospital or health center. In the case of a sickle cell crisis, 56.4% of the caregivers preferred to take the patients to health centers, compared to 25.7% who preferred to visit SMC. About 15.8% used traditional medicine or simple analgesics, and 6.9% went to private clinics to manage the painful sickle cell crisis. When caregivers were asked to choose a reason for selecting a particular medical facility as the first destination, most caregivers who preferred going to the health centers agreed that this was due to its proximity to their houses and the good quality of the services. Conversely, those who preferred going to SMC were more satisfied with the quality of services. In addition, some participants complained about the delay in the accidents and emergency department of SMC, which was why they preferred other facilities with a faster response than SMC (Fig 1).

Among the caregivers of patients with SCD, 73.3% had high family support, and about 81.2% of the support was emotional support, which they described as listening and showing empathy. Other types of support the participants said they received were financial (18.8%) and

**Table 1. Sociodemographic characteristics of the caregivers and patients with SCD in Bahrain (n = 101).**

| *Caregivers* | | N |
|---|---|---|
| Age (Years) | < 40 | 18 |
| | 40–50 | 67 |
| | > 50 | 15 |
| Gender | Male | 23 |
| | Female | 78 |
| Marital status | Single | 2 |
| | Married | 93 |
| | Divorced | 2 |
| | Widowed | 4 |
| Relationship with patient | Mother | 74 |
| | Father | 23 |
| | Other relatives | 4 |
| Current occupation | Employed | 52 |
| | Unemployed | 49 |
| Educational level | Illiterate | 1 |
| | Primary | 3 |
| | Intermediate | 15 |
| | High school | 41 |
| | Diploma | 13 |
| | Bachelor | 20 |
| | Master | 7 |
| *Patients* | | |
| Age (Years) | 10–12 | 33 |
| | 13–15 | 42 |
| | 16–18 | 26 |
| Gender | Male | 55 |
| | Female | 46 |

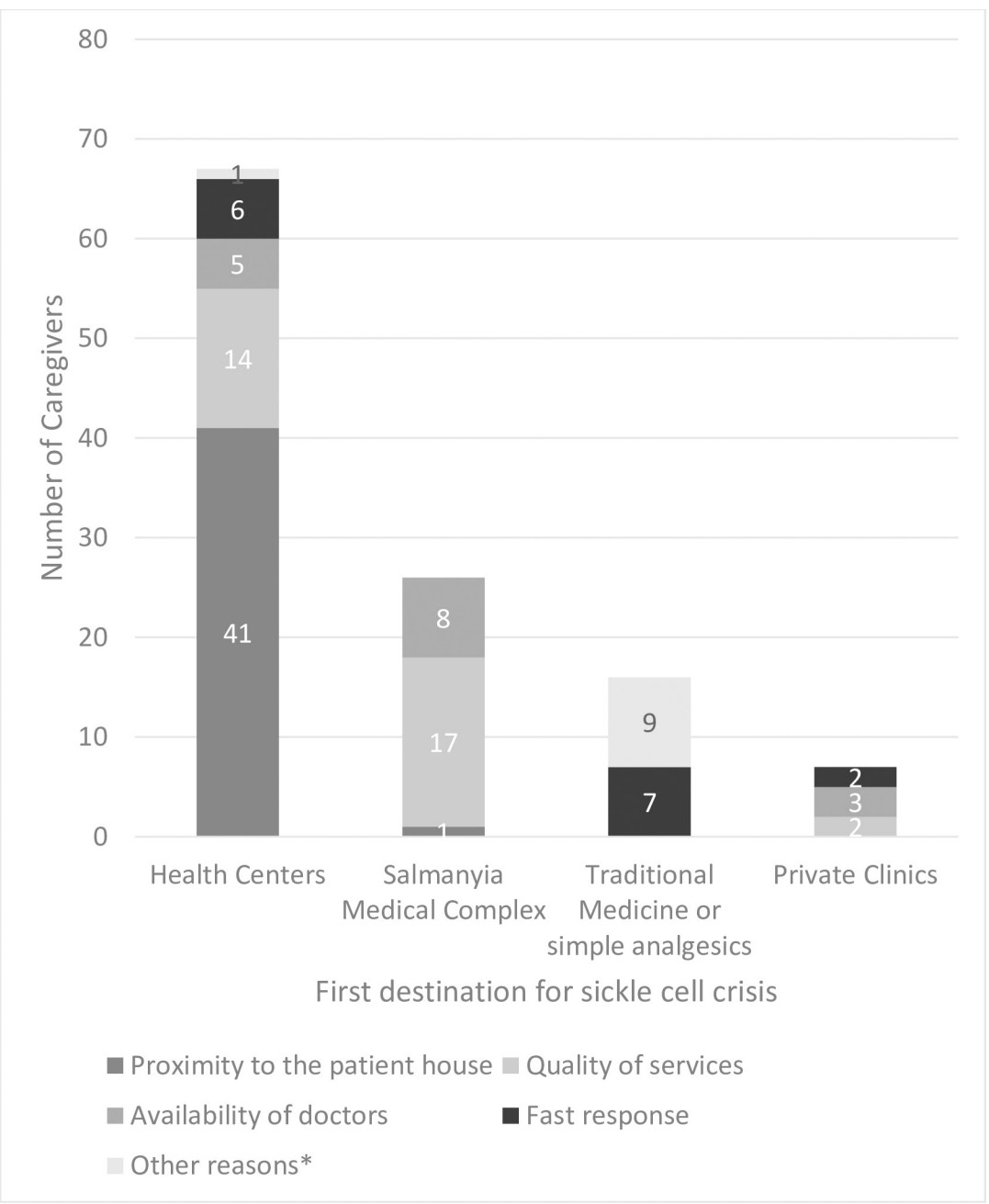

**Fig 1. Caregivers' reasons for selecting various types of health facilities as the first destination for a sickle cell crisis.** (*
Other reasons: avoid admission, mild crisis).

informational support (23.8%), and 4% of caregivers received other forms of support from their families in which other family members accompanied the affected child to a medical appointment or hospital stay. For most caregivers (73%), caring for an adolescent with SCD had a low impact on their relationships with others. Although 48.5% of caregivers stated that their time was moderately affected because of the time they spent with their children with SCD, 39.6% complained that their time was extremely affected. Regarding the financial aspect, we found that 14.8% of the caregivers had catastrophic health expenditures as they spent more

than 10% of their total family income on their patients' health. More than half of the caregivers who suffered catastrophic health expenditures reported that other family expenses were also affected. The mean out-of-pocket expenditure on SCD care was 16.72 Bahraini Dinar (1 Bahraini Dinar = 2.65 USD) per month with a range from 0 to 150 Bahraini Dinars. The mean monthly household was 776 Bahraini Dinar, with a median of 500 Bahraini Dinar.

## Qualitative results

Most of the Bahraini caregivers were in their fourth and fifth decades of life, and seventeen out of eighteen interviews were with females. Three distinct and interrelated themes were identified in our analysis: A) The intricacy of caring for Patients with SCD which includes the sub-themes: the psychological tragedy, caregiving hardships, and cost of care on families. B) Dissatisfaction with hospital facilities C) Insufficient healthcare services. Fig 2 shows a summary of the coding process, subthemes, and themes.

**1. The intricacy of caring for SCD patients.**  *Subtheme A*: *Psychological tragedy*. The moment upon receiving the SCD diagnosis was considered a tragedy by almost all caregivers. "Shock," "breaking down," "collapsing," "crying," and "sadness" were some emotions used to describe this event. Caregivers often thought about their children's fate with the disease.

*"In the beginning, I was shocked; I could not even sleep. I felt the news was surprising, and I was surprised. It felt like a dream, like someone told me in a dream, and I want to wake up from it." (Mother of a 17-year-old female, aged 42 years, IDI-11)*

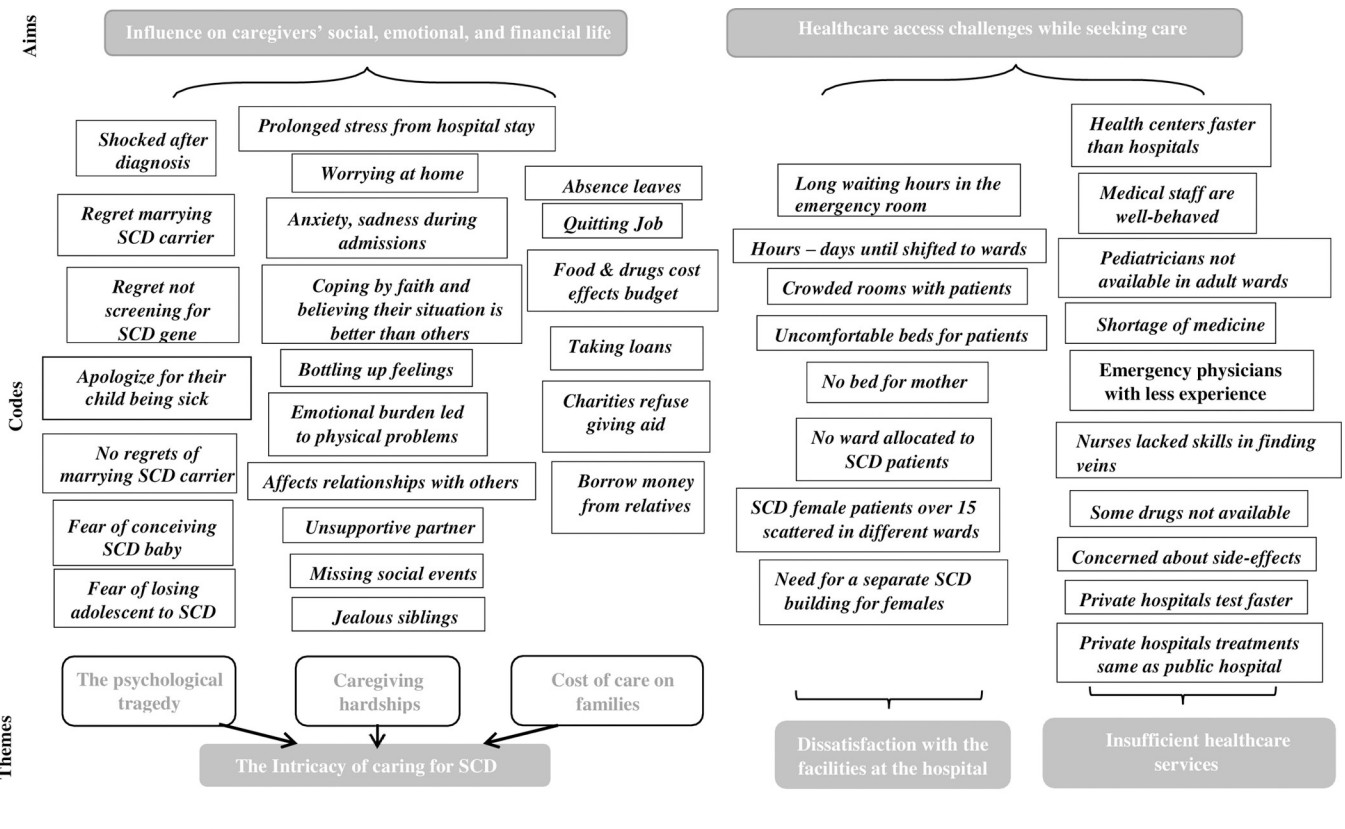

SCD = Sickle Cell Disease

**Fig 2. Summary of the coding process, subthemes, and themes.**

Many caregivers showed intense feelings of guilt and regret regarding marrying a carrier husband with the genetic mutation for SCD or not having performed the genetic screening for SCD prior to marriage. They often blamed themselves for their children's pain, and some constantly apologized to their children for not being able to enjoy a normal childhood. However, some mothers had no regrets because it was not their choice, and the premarital test was not mandatory.

*"I cry a lot because I am in pain and regret. You cannot even imagine how much I blame myself." (Mother of a 10-year-old male, aged 38 years, IDI-1)*

*"I regret getting married to a sickle cell carrier; these kids are killing me inside. If I could go back in time, I would not get married to the same person." (Mother of an 18-year-old female, aged 43 years, IDI-8)*

*"In the beginning, I told her, 'I apologize for bringing you to this life, and you are the only one who is sick in this house. You did not have a normal life or normal childhood.' And when she got older, she did not get the chance to have a life and play like the other kids at her age." (Mother of an 18-year-old female, aged 43 years, IDI-8)*

Several caregivers discussed their fear of conceiving and the chance of having another child with SCD, as well as their fear of losing their child to the disease or its complications.

*"I became pregnant with my second daughter; I was too scared that she may also be born with the disease." (Mother of a 16-year-old female, aged 47 years, IDI-18)*

*"Once she (the child) gets sick I always say I will lose her, because of the stories I have heard about death among Patients with SCD, although her attacks are mild. But once she gets an attack, I get too scared that I might lose her."(Mother of a 14-year-old female, aged 38 years, IDI-4)*

*Subtheme B*: *Caregiving hardships*. After the initial shock, regret, and guilt at diagnosis, caregivers underwent struggles regarding acceptance and adaptation. Most caregivers faced psychological stress due to prolonged hospital stays during their children's admissions. Caregivers discussed how they would spend sleepless nights in hospitals and at home, worrying about their child's healing process. They felt chronically fatigued, anxious, and sad. However, caregivers' faith enhanced their coping ability and acceptance of their situation to be better than the circumstances of other people.

*"I am psychologically in pain. I am tired; I am tired." (Mother of a 14-year-old female, aged 38 years IDI-4)*

*"In the hospital, he meets a lot of patients with worse medical conditions. Allah loves you and that is why he gifted you with this disease; look at the people around you who cannot walk or hear, and some of them have even lost their parents. Allah gave you something and took something from you; you should thank Allah for all the blessings that you have." (Mother of a 10-year-old male, aged 34 years, IDI-16)*

Some of the caregivers said they would not openly talk with other people about their emotional burden. They always acted like everything was fine in front of their families to avoid creating a sad environment.

*Researcher: "During an emotional breakdown, with whom do you share your feelings and open up about it?" Mother: "Honestly, I cry alone." (Mother of a 17-year-old male, aged 40 years, IDI-9)*

This emotional burden, psychological stress, overthinking, and insomnia often lead to physical problems in caregivers. They complained of several health issues, including gastrointestinal and cardiovascular symptoms.

*"I suffer from some health issues. I suffer from spasms and stomach pain, and it increases with my anxiety when I care for her. I make sure that I will not make her aware of this; the spasms increase a lot while I am worried and angry." (Mother of an 18-year-old female, aged 43 years, IDI-8)*

Although most mothers stated that they had support from their husbands in caring for the child, some caregivers mentioned that their relationship with their spouses was affected since the diagnosis of their child with SCD because they blamed each other for their child's suffering. Some mothers also indicated that their husbands did not spend enough time with their children during the crisis, and some could not bear to see their children in pain.

*"He (the husband) told my family that I have changed since I became pregnant with my daughter, and I changed more once I knew that she is sick with sickle cell disease, since then he has changed too." (Mother of an 18-year-old female, aged 43 years, IDI-8)*

Given that caregivers spent most of their time accompanying their affected child during the crisis, caregivers stated that they were missing socializing with their family, friends, and special events. At home, other siblings felt jealous and neglected because of the extra attention the child with SCD received.

*"I used to go out with my husband's family, and my friends, but now (after having the affected child) I have disappeared; no one sees me!" (Mother of a 12-year-old male, aged 40 years, IDI-10)*

*"He (sibling) says, 'You left us alone and went to her (affected child).' He feels jealous of her." (Mother of a 17-year-old female, aged 41 years, IDI-11)*

*Subtheme C*: *Cost of care on families*. Some caregivers took numerous leaves of absence or even sacrificed their careers to stay with the affected child in the hospital.

*"I used to work, but now I resigned to take care of my girl." (Mother of a 16-year-old female, aged 43 years, IDI-18)*

Participants also reported that their expenses increased when their children were hospitalized. Most caregivers said that buying food and medications during hospital admissions significantly affected their budgets.

*"To be honest, the food is extremely costly for me, because he does not like the hospital's food." (Mother of a 10-year-old male, aged 38 years, IDI-1)*

Caregivers with poor financial backgrounds coped with increased expenses during their child's hospitalization through several methods, including loans and monthly financial support from

charities. Two caregivers mentioned that despite their urgent needs, a charity refused to provide a stipend under the pretext that there were more deserving families. Caregivers discussed how embarrassing it was for them and their husbands to borrow money from their relatives.

*"We are living on 150 BD (Bahraini Dinar)." (Mother of a 12-year-old male, aged 40 years, IDI-10)*

*"We put so much pressure on ourselves and borrowed money from my husband's sister, but he does not like to ask her; it is so hard on men to do such a thing." (Mother of a 16-year-old female, aged 43 years, IDI-18)*

**2. Dissatisfaction with hospital facilities.** Most caregivers discussed experiencing delays in the emergency room. They waited several hours before receiving treatment. In addition, they had to wait several hours and sometimes days in the emergency room until the patients were shifted to the appropriate wards.

*"It takes us hours until we are referred to the emergency room, and then we also wait hours until they come to give him (the child) the medicine." (Mother of a 12-year-old male, aged 40 years, IDI-10)*

*"My daughter waited for one day in the emergency room; they did not refer her to the ward for two days." (Mother of 17-years-old female, aged 42 years, IDI-11)*

Most participants experienced physical fatigue and back pain because of the lack of space for the accompanying mothers in the pediatric wards, leading them to sleep on the floor for as long as two to five weeks.

*"Just imagine, the mother sleeps on the chair or on the floor and not for a day or two; they stay like this for about three to five weeks. It is extremely hard; with all this, it is not surprising to get physically fatigued." (Mother of a 10-year-old male, aged 34 years, IDI-16)*

In addition, some caregivers stated that the patients' beds in the hospital were uncomfortable and even worsened the pain of the adolescent.

*"My daughter enters the hospital, for example due to pain in her arm, but the pain radiates to her back; she (my daughter) said because of the beds, they are uncomfortable." (Mother of an 18-year-old female, aged 43 years, IDI-8)*

According to many caregivers, the space designated for patients with SCD in the main tertiary care hospital was scarce and hardly accommodated a large number of patients. Therefore, wards and emergency rooms were usually crowded with patients and their accompanying relatives, creating an uncomfortable atmosphere.

*"The space for patients with SCD in the hospital is very, very small; it does not accommodate all this number (of patients)." (Mother of an 18-year-old female, aged 43 years, IDI-8)*

Another problem identified by most caregivers was the lack of a separate ward in the hospital dedicated to patients with SCD. In addition, some patients wished for a separate building for female patients with SCD older than 15 years, because they were currently scattered within different wards, while male patients with SCD had their separate building.

*"Males with sickle cell disease have a separate building but females do not; until when female's patients will suffer from this?" (Mother of an 18-year-old female, aged 43 years, IDI-8)*

**3. Insufficient healthcare services.** Caregivers preferred to visit health centers over hospitals because of the previous unpleasant experiences of other family members in the large hospital on several occasions. For example, the availability of medication and response to patients was much better in health centers than in hospitals.

*"In the health center, the care is much better. They let us in as we arrive, bring the doctor, and treat him (the patient), but in hospitals I have to wait for three hours (before we receive the treatment)." (Mother of a 12-year-old male, aged 39 years, IDI-12)*

Many caregivers mentioned that hospital staff are respectful, have good behavior, and are always welcoming to the patients. They also mentioned that doctors and nurses were humble and treated their patients as if they were part of their own families.

*"Honestly, I cannot say anything bad about them; they try their best with my son as if they're his parent." (Mother of a 10-year-old male, aged 38 years, IDI-1)*

However, many caregivers reported that one of the main issues that their children faced was the need to be followed up by a pediatrician. Currently, they are transferred to an adult ward after they turn 15 years old, thus pediatricians are not always available to them in those wards. Moreover, caregivers stated that the existing number of doctors and nurses in the emergency room did not meet the patient's needs.

*"It's the emergency room! There should have been more doctors and nurses; one or two will not be enough for a long waiting line." (Mother of a 17-year-old female, aged 42 years, IDI-11)*

Caregivers believed some health workers who required more experience in performing minor procedures. They discussed that some nurses lacked skill in finding veins to inject medicines, leading to irritation, pain, and anxiety in their children, and that emergency room physicians required more experience in treating patients with SCD.

*"Nurses were joyful with my child; they were nice, but while taking blood, my child was screaming." (Mother of a 17-year-old male, aged 40 years, IDI-9)*

Caregivers stated that most prescriptions were available in hospitals. However, some medicines, such as diclofenac sodium, were usually unavailable, so they had to buy them out of pocket. In addition, caregivers were hesitant about using hydroxyurea as the drug of choice for their children because of a wide range of side effects, including infertility, based on information received from their relatives and other acquaintances. However, medical staff counseled them to alleviate their doubts and concerns.

*"I was scared once I gave her the (Hydroxyurea) capsules; people would tell me that it will lead to hair loss and problems in getting pregnant. "(mother of a 10-year-old female, aged 50 years, IDI-13).*

Some of the caregivers stated that laboratory tests in private hospitals were faster than those in public hospitals and that they preferred going to private rather than public hospitals for laboratory tests. Other caregivers stated that private clinics did not differ much from public hospitals and that the treatment was equal in both types of hospitals.

*"Sometimes, I prefer to go to private hospitals to get the blood tests done, because the public hospitals take time, and maybe he will not need to stay in hospital so I just skip the seven or eight hours waiting (in the emergency room) and (directly) go to a private hospital." (Mother of a 10-year-old male, aged 34 years, IDI-16)*

## Discussion

This mixed-methods study aimed to investigate the health care access problems faced by caregivers of patients with SCD and the impact of being a caregiver on various aspects of their lives. Difficulties with parking in the hospital and prolonged waiting in the emergency room discouraged more than half of the caregivers who preferred to seek care from smaller healthcare centers. The lack of a separate ward for patients with SCD and dissatisfaction with some of the facilities and services at the major tertiary care hospital were some of the major challenges these caregivers experienced while seeking care for their children. Significant emotional and financial burdens were accompanied by fear and regrets about their children, which could be associated with stress and physical symptoms among caregivers. The most pertinent results are presented below.

Caregiving had a massive impact on the caregivers. Many were afraid of losing their children due to the complications of the disease and worried about having another child born with SCD. It was also common for caregivers to stay in the hospital for long periods while their children were hospitalized, which negatively affected their psychological health. More than one-third of the participants in the qualitative study complained of psychological stress, which sometimes deteriorated their physiological and mental well-being. These results are consistent with a cross-sectional study from Saudi Arabia, which showed that emotional health is one of the most affected aspects of caregivers' quality of life [41], and at least four studies from Africa, where the burden of SCD is also the highest in the world, that show a high psychological burden on the caregivers of patients with SCD [20, 42–44].

Regret was another common feeling among the participants. Some caregivers regretted marrying a carrier husband, whom they believed was the reason for their children's pain. Other caregivers experienced no regret because the premarital test was not mandatory when they were married. Despite the great emotional burden on caregivers, quantitative analysis showed that 81.2% of them received emotional support from their families. This emotional support is necessary for primary caregivers whose quality of life is often affected by the long-term care that they provide their sick children [18].

Although the quantitative analysis in this study showed that 73.3% of caregivers stated that caring for sick children did not greatly affect their relationships with partners, children, relatives, friends, and coworkers, the qualitative findings differed. The qualitative results showed that the social lives of caregivers seemed to have been drastically affected since they were scared to leave their children without their attention. The differing quantitative and qualitative results may have been because the caregivers freely talked about their challenges in an in-depth discussion for the latter. Many socially active caregivers stopped participating in their family and friends' gatherings to look after their children. Even siblings of the patients felt jealous or neglected unintentionally because of the extra attention the caregivers gave the affected

child. The current findings are consistent with studies from the United States and Saudi Arabia, where caregivers of SCD reported neglecting other family members and spending less time with them because of caring for their affected children [22, 41].

Financial burden was variable among caregivers. Many had no financial problems, while others indicated their poor financial status, which worsened when the child experienced a sickle cell crisis and was hospitalized. Caregivers gave several reasons for their worsening financial status. First, the expenses increased during admissions since they must buy food because most children would not like hospital food, while others bought medications not usually provided by the hospital. Second, some caregivers had to be absent several times or even leave their jobs to take care of their children, which impacted their earnings [45]. These results are consistent with a study from Nigeria that showed that 56.8% of SCD caregivers lost between 1 and 48 working days taking care of their affected child. Three caregivers lost their jobs for the same reason [23]. The quantitative analysis of the data in this study also showed that 14.8% of caregivers suffered catastrophic health expenditures, which is considered relatively high based on the free healthcare services in Bahrain. The catastrophic health expenditure caring for patients with SCD can be as high as 20.7% in some countries [23].

Patients with SCD require frequent visits to the emergency or outpatient departments because of persistent symptoms, which may sometimes be difficult to treat [46]. However, visits to EDs may not be smooth. Long waiting hours in the emergency room were a major problem faced by most caregivers, as supported by the qualitative analysis in this study. As a result, caregivers often preferred home management with different analgesics for pain relief during sickle cell crises before considering seeking healthcare services. The delay in the emergency room also determined their preference for either management at home or from primary healthcare centers during a sickle cell crisis. Some caregivers stated that there was a shortage of staff in the emergency room, which the researchers believe might be associated with delays in providing the treatment. Studies from the USA have shown that patients with SCD experience significant delays in receiving treatment in emergency departments [32, 33].

Another major problem in accessing health care was that patients with SCD, especially women older than 15 years, did not have a dedicated ward. They were scattered among different adult wards assigned to diseases other than SCD, even though they are still treated by pediatricians, who are not always available in adult wards, during their hospitalization, making it difficult for caregivers to know about their children's condition. In addition, many patients disliked going to the hospital during crises because they could not have a good relationship with adult patients, unlike when they were in pediatric wards where most of the patients were around their age. An innovative and culturally appropriate solution may be implemented with the involvement of a multidisciplinary team to enhance satisfaction and health outcomes in patients with SCD [47].

## Strengths and limitations

The main strength of the current study was its mixed-method study design, consisting of qualitative and quantitative components that complemented each other. This study had several limitations; hence, the findings should be interpreted with greater caution. Although the study's questionnaire underwent a pilot study and was tested for validity and reliability within the study population, missing data reduced the sample size for out-of-pocket expenses from 101 to 88. This was mitigated during the analysis by omitting incomplete data. However, it represents a randomized 15% (88/571) of the estimated population, which is a good representation. Larger sample sizes in future studies could yield more precise estimates. Caregivers reported events and expenses at different periods, which may introduce recall bias. However, the

researchers attempted to help them in their recall efforts. The study only covered SCD caregivers who used services from a single tertiary hospital in Bahrain; therefore, the findings may not be generalizable. However, this hospital is the largest tertiary care hospital in Bahrain and provides healthcare services to the entire population of Bahrain. Another limitation of this study was that the participants in the qualitative portion were mainly caregivers of hospitalized patients, while most of the participants in the quantitative part were caregivers of patients from outpatient clinics. Hospitalized patients usually have moderate-to-severe disease, while those in outpatient clinics mostly have mild disease. This may also explain why the results from some aspects of this study are not consistent with others, such as the impact on the relationship of caregivers where the qualitative and quantitative analyses revealed high and low impacts, respectively. Another limitation is that the qualitative and quantitative studies were conducted at the same time; therefore, many unexpected new findings not identified by the quantitative questionnaire were revealed by the qualitative analysis. This can be explored in future research; the items from the qualitative analysis in this study could be included in a more robust quantitative study identifying the gaps in service delivery to meet the expectations of the caregivers and their patients. The perspective of the providers and administrators in the hospital, which may be different from those of the patients, was not sought in this study. There could be operational reasons for delays in the emergency room that must be explored further.

## Conclusion

Caregivers' lives were mainly affected by their adolescent's chronic medical illness, such that they experienced emotional burdens involving fear, regrets, and guilt. These ongoing psychological stressors would impact caregivers' quality of life and care of individuals with SCD, increasing the healthcare burden. Thus, it is vital to overcome barriers to the healthcare system by facilitating easy access to the facilities, improving service delivery issues, such as long waiting times in the emergency room, and providing a dedicated hospitalization space for female adolescents with SCD. This research overshadows some productive insights on the importance of advocacy for better healthcare policies related to SCD. These findings should help to improve access to appropriate services, reduce the healthcare burden, and lower the cost of healthcare expenditures on families with an adolescent with SCD. Formal support is essential by creating organizations or social communities that can help to provide social, emotional, and financial support to families with an adolescent with SCD [4, 23, 35, 41]. A recent study suggests that telemedicine was a useful tool to enhance access to service and shared caregivers' experiences with children with SCD following the pandemic of COVID-19 [48].

## Supporting information

**S1 Data. Software data.**
(SAV)

**S2 Data. Data output.**
(PDF)

**S3 Data. Qualitative data analysis.**
(DOCX)

**S1 Appendix. Survey questionnaire in English.**
(DOCX)

**S2 Appendix. In-depth interview guide in English.**
(DOCX)

**S3 Appendix. Survey questionnaire in Arabic.**
(DOCX)

**S4 Appendix. In-depth interview guide in Arabic.**
(DOCX)

## Acknowledgments

We are grateful to the parents of the patients with SCD who participated in our research. We are also grateful to the nurses in the pediatric ward and outpatient department of the SMC who helped us find the participants. We thank Mr. Amer AlMarabeh, a biostatistician at the Arabian Gulf University, for his help with quantitative data analysis.

## Author Contributions

**Conceptualization:** Khadija Al Saif, Fatema Mohamed Abdulla, Anwaar Alrahim, Sara Abduljawad, Zainab Matrook, Jenan Jaafar Abdulla, Fatima Bughamar, Fatema Alasfoor, Rana Taqi, Amna Almarzooq.

**Data curation:** Khadija Al Saif, Anwaar Alrahim, Zainab Matrook.

**Formal analysis:** Khadija Al Saif, Fatema Mohamed Abdulla, Anwaar Alrahim, Sara Abduljawad, Zainab Matrook, Jenan Jaafar Abdulla, Fatima Bughamar, Rana Taqi, Amna Almarzooq.

**Funding acquisition:** Khadija Al Saif, Fatema Mohamed Abdulla, Anwaar Alrahim, Sara Abduljawad, Zainab Matrook, Jenan Jaafar Abdulla, Fatima Bughamar, Fatema Alasfoor, Rana Taqi, Amna Almarzooq.

**Investigation:** Khadija Al Saif, Fatema Mohamed Abdulla, Anwaar Alrahim, Sara Abduljawad, Zainab Matrook, Jenan Jaafar Abdulla, Fatima Bughamar, Fatema Alasfoor, Rana Taqi, Amna Almarzooq.

**Methodology:** Khadija Al Saif.

**Project administration:** Khadija Al Saif, Zainab Matrook.

**Resources:** Khadija Al Saif.

**Software:** Khadija Al Saif, Anwaar Alrahim.

**Supervision:** Jamil Ahmed.

**Validation:** Khadija Al Saif.

**Visualization:** Khadija Al Saif, Sara Abduljawad, Fatema Alasfoor, Jamil Ahmed.

**Writing – original draft:** Khadija Al Saif, Fatema Mohamed Abdulla, Anwaar Alrahim, Sara Abduljawad, Zainab Matrook, Jenan Jaafar Abdulla, Fatima Bughamar, Fatema Alasfoor, Rana Taqi, Jamil Ahmed.

**Writing – review & editing:** Khadija Al Saif, Jamil Ahmed.

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
