## [Decision Letter · Decision Letter 0]

15 May 2021

PONE-D-20-34450

The Impact of Being a Caregiver, and Barriers to Seeking Care for Adolescents with Sickle Cell Disease in Bahrain

PLOS ONE

Dear Dr. Ahmed,

Thank you for submitting your manuscript to PLOS ONE. After careful consideration, we feel that it has merit but does not fully meet PLOS ONE’s publication criteria as it currently stands. Therefore, we invite you to submit a revised version of the manuscript that addresses the points raised during the review process.

We look forward to receiving your revised manuscript.

Kind regards,

Rachel A. Annunziato, Ph.D.

Academic Editor

PLOS ONE

Journal Requirements:

2. Thank you for providing the English version of the questionnaires. Please also include a copy of both questionnaires in the original language, as Supporting Information, or include a citation if it has been published previously.

3. For the quantitative study, please provide further details on sample size and power calculations.

6. Please upload a copy of Figure 1 and 2, to which you refer in your text on page 12 and 13. If the figure is no longer to be included as part of the submission please remove all reference to it within the text. We note that Figure 1 and 2 may be duplicate of your Supplementary Figures.

7. Please include captions for ALL your Supporting Information files at the end of your manuscript, and update any in-text citations to match accordingly. Please see our Supporting Information guidelines for more information: http://journals.plos.org/plosone/s/supporting-information.

Reviewers' comments:

Reviewer's Responses to Questions

**Comments to the Author**

1. Is the manuscript technically sound, and do the data support the conclusions?

Reviewer #1: Yes

Reviewer #2: Partly

2. Has the statistical analysis been performed appropriately and rigorously? 

Reviewer #1: Yes

Reviewer #2: Yes

3. Have the authors made all data underlying the findings in their manuscript fully available?

Reviewer #1: Yes

Reviewer #2: Yes

4. Is the manuscript presented in an intelligible fashion and written in standard English?

Reviewer #1: Yes

Reviewer #2: No

5. Review Comments to the Author

Reviewer #1: This is in an important study with the aim of determining the caregivers’ difficulties in seeking care for their patients with sickle cell disease. The view of the caregiver is often ignored when research is conducted into chronic diseases/disorders like sickle cell disease.

1. The authors need to thoroughly edit their work, few spelling errors are highlighted in the manuscript.

2. Other comments are included in the manuscript

Reviewer #2: Comments

This study describes the impacts of caregiving among relatives of adolescents with SCD and the barriers to care in a Bahraini tertiary hospital. Importantly, the study provides important perspectives on the experience of caregivers of adolescents with SCD in Bahrain. However, the paper needs substantial improvement in the style of writing and correction of grammatical/typo errors.

Title:

Given that the study was conducted in a single tertiary hospital, it is much better to indicate this in the title unless there is clear justification that the study sample is representative of national population of Bahrain.

The paper needs English Language editing and revisions of typo errors

Abstract

• Typographical errors should be corrected. For instance, the authors are encouraged to revise the first sentence in the abstract:

” We aim to determine the perspective of caregivers on the difficulties encountered while seeking care for their adolescents with sickle cell disease (SCD)”.

• The authors are encouraged to use first person language throughout the paper. For example, the phrase: “SCD patients” should be changed to “adolescents with SCD”

• The sentence: “In the quantitative part of the study, we used……...” needs revision to improve clarity.

• The rationale for an emphasis on “Salmaniya Medical Complex” here is not entirely clear.

• The authors are encouraged to revise the sentence: “the impact of caring for SCD patients, dissatisfaction with the facilities at the hospitals, services provided during hospital visits were not enough”. This sentence is not clear.

Introduction:

• The second sentence in the introduction alluded that the prevalence of SCD in the Middle Eastern region is driven by “large family size”. It is important for the author to provide the scientific basis for making this conclusion.

• The introduction can be improved by operationalizing the main study outcome(s), and describing the study objectives/postulations/or hypotheses.

Methods:

• The rationale for the sample size should be included with citations, if available. If the estimation of the sample size or power calculation is not feasible, this should be included in the study limitations.

• The authors are encouraged to confirm informed consent/assent from caregivers versus adolescents.

• It will be nice to include information on the study response rate.

• A copy of the study questionnaire should be provided as supplementary material.

• The layout of the methods needs improvement to ensure clarity. For example, the quantitative and qualitative components of the study can be described side by side in a logical and clear fashion. This will help avoid the creation of two subsections on data-analysis

Results

• Revisions to correct typos and all grammatical errors

• The sentence: “Three distinct but ……………….” should have “and’’ connecting the third reasons listed.

• The authors are encouraged to revise the presentation of the data collected from the qualitative assessment. The results of the qualitative data are currently difficult to follow and to lengthy

Table legend should be provided to describe all abbreviations used in the table

Discussion

• The use of the phrase, “catastrophic health expenditures” sounds somewhat awkward.

• In the absence of a clearer presentation of the shortcomings associated with receiving care from health centers that are closer to the residence of the study participants residence, the caregiver’s preference in attending these centers for treatment compared to the tertiary center seems reasonable.

• The cross-sectional study design, a lack of hypothesis-driven sample size estimations or power calculation, and data was collected in one tertiary hospital limit the generalization of the study findings. This should be included the study shortcomings.

6. PLOS authors have the option to publish the peer review history of their article (what does this mean?). If published, this will include your full peer review and any attached files.

Reviewer #1: No

Reviewer #2: **Yes: **Andrew T. Olagunju

---

## [Author Response · Author response to Decision Letter 0]

3 Sep 2021

Reference number: PONE-D-20-34450

Dear Editor Rachel A. Annunziato,

Thank you for giving us the opportunity to submit a revised draft of our manuscript titled [Barriers to Seeking Care for Adolescents with Sickle Cell Disease in a Tertiary Care Hospital in Bahrain: Caregiver Experiences] to PLOS ONE. We appreciate the time and effort that you and the reviewers have dedicated to providing your valuable feedback on our manuscript. We are grateful to the reviewers for their generous comments on our paper. We have been able to incorporate changes to reflect most of the suggestions provided by the reviewers and revised the manuscript accordingly. We have highlighted the changes within the manuscript. Our responses to these comments are given under each of your comment. 

and

Response to editor: Thank you for reminding us about it. We have made sure to follow the journal guidelines. We reviewed the style requirements and made the necessary changes.

2. Thank you for providing the English version of the questionnaires. Please also include a copy of both questionnaires in the original language, as Supporting Information, or include a citation if it has been published previously.

Response to editor: We added both questionnaires in the original language as supporting information.

3. For the quantitative study, please provide further details on sample size and power calculations.

Response to editor: We provided details on the sample size and power calculations in the methods section with citations.

Response to editor: Given your concern, we provided the analysis process for both the quantitative and qualitive methods in the supporting information files. This is the minimal anonymized data set required to replicate our study findings.

Response to editor: We revised the full manuscript to ensure that the ethics statements appear only in the Methods section.

6. Please upload a copy of Figure 1 and 2, to which you refer in your text on page 12 and 13. If the figure is no longer to be included as part of the submission please remove all reference to it within the text. We note that Figure 1 and 2 may be duplicate of your Supplementary Figures.

Response to editor: We uploaded a copy of Figures 1 and 2 from our text.

7. Please include captions for ALL your Supporting Information files at the end of your manuscript, and update any in-text citations to match accordingly. Please see our Supporting Information guidelines for more information: http://journals.plos.org/plosone/s/supporting-information

Response to editor: We included captions for all our supporting information files at the end of the manuscript. We also updated the in-text citations to match the supporting information file.

Here is a point-by-point response to the reviewers’ comments and concerns.

Comments from Reviewer #1:

• Comment 1: This is in an important study with the aim of determining the caregivers’ difficulties in seeking care for their patients with sickle cell disease. The view of the caregiver is often ignored when research is conducted into chronic diseases/disorders like sickle cell disease. The authors need to thoroughly edit their work, few spelling errors are highlighted in the manuscript.

• Response to reviewer: Thank you for your comments. We reread the paper carefully and hired a professional to correct all the grammatical errors and provide English editing.

Other comments are included in the manuscript:

• Comment 2: The authors should consider rephrasing this sentence, it currently read as if it’s the care giver who is receiving treatment.

• Response to reviewer: We rephrased the sentence to clarify the meaning.

• Comment 3: This should be cited appropriately either as personal communication or unpublished research.

• Response to reviewer: We agree with this comment. We do not have any statistics available that give us the exact number of adolescents with SCD aged 10-18 years in Bahrain during 2019. However, we cited the number of our target population by the recorded number of newborns with SCD in published research. For the rest of our target population, we mentioned the published number of students with SCD in each academic school year in newspapers.

• Comment 4: In the event that there were incomplete data, what did the data collectors do?

• Response to reviewer: We did not use questionnaires that did not fulfill the purpose and objectives of the study. Questions that caregivers could not fill out because the answers differed at various times and questions that were sensitive to the caregiver were omitted from the calculations, though we did include these caregivers in the study.

• Comment 5: Translated by who? Double checked by who?

• Response to reviewer: It was translated and double checked by the researchers. We included their initials in the manuscript. 

• Comment 6: How were these participants selected from the larger group of 101 caregivers?

• Response to reviewer: Out of 101 caregivers in the quantitative study, only two caregivers were included in the qualitative study randomly. We conducted the two studies separately and we did not select participants from other part of the study.

• Comment 7: According to WHO adolescence starts from the age of 10. Why did the authors choose to label this group adolescent?

• Response to reviewer: Thank you for noticing the typo. We meant to separate adolescence into early adolescence (10-14 years of age) and late adolescence (15-18 years of age).

• Comment 8: The age of some caregivers is not given, this should be corrected to make it uniform.

• Response to reviewer: We provided the quotes with the caregivers’ ages.

• Comment 9: ‘absence’?

• Response to reviewer: The sentence has been changed to absence.

• Comment 10: This was not a finding in this particular study.

• Response to reviewer: Thank you for noticing this statement. We modified it with the proper studies.

Comments from Reviewer #2:

• Comment 1: This study describes the impacts of caregiving among relatives of adolescents with SCD and the barriers to care in a Bahraini tertiary hospital. Importantly, the study provides important perspectives on the experience of caregivers of adolescents with SCD in Bahrain. However, the paper needs substantial improvement in the style of writing and correction of grammatical/typo errors.

• Response to reviewer: Thank you for your time and effort in reviewing our article. This review helped us improve the quality and value of the article. We edited our work and corrected the errors.

Title:

• Comment 2: Given that the study was conducted in a single tertiary hospital, it is much better to indicate this in the title unless there is clear justification that the study sample is representative of national population of Bahrain.

• Response to reviewer Thank you, we have updated the title. We revised the title to be more concise.

• Comment 3: The paper needs English Language editing and revisions of typo errors

• Response to reviewer: We hired a professional editor to review and improve the language of the article.

Abstract:

• Comment 4: Typographical errors should be corrected. For instance, the authors are encouraged to revise the first sentence in the abstract:

” We aim to determine the perspective of caregivers on the difficulties encountered while seeking care for their adolescents with sickle cell disease (SCD)”.

• Response to reviewer: We revised the abstract and changed this sentence.

• Comment 5: The authors are encouraged to use first person language throughout the paper. For example, the phrase: “SCD patients” should be changed to “adolescents with SCD”

• Response to reviewer: We changed the wording “SCD patients” to “adolescents with SCD” when appropriate throughout the paper.

• Comment 6: The sentence: “In the quantitative part of the study, we used……...” needs revision to improve clarity.

• Response to reviewer: We revised this sentence.

• Comment 7: The rationale for an emphasis on “Salmaniya Medical Complex” here is not entirely clear.

• Response to reviewer: Thank you for pointing this out. We removed this sentence from the abstract.

• Comment 8: The authors are encouraged to revise the sentence: “the impact of caring for SCD patients, dissatisfaction with the facilities at the hospitals, services provided during hospital visits were not enough”. This sentence is not clear.

• Response to reviewer: We modified the themes and explained them in the abstract.

Introduction:

• Comment 9: The second sentence in the introduction alluded that the prevalence of SCD in the Middle Eastern region is driven by “large family size”. It is important for the author to provide the scientific basis for making this conclusion.

• Response to reviewer: We removed this text as we did not find a scientific basis for this conclusion.

• Comment 10: The introduction can be improved by operationalizing the main study outcome(s), and describing the study objectives/postulations/or hypotheses.

• Response to reviewer: We revised the Introduction. We tried our best to find literature related to the main study outcomes. We expanded the last paragraph of the Introduction to provide more clarity on the objectives/rationale of the study.

Methods:

• Comment 11: The rationale for the sample size should be included with citations, if available. If the estimation of the sample size or power calculation is not feasible, this should be included in the study limitations.

• Response to reviewer: Thank you for this thoughtful suggestion. We have provided the sample size estimation and power calculations, with citations.

• Comment 12: The authors are encouraged to confirm informed consent/assent from caregivers versus adolescents.

• Response to reviewer: Thank you for pointing this out. We clarified this statement.

• Comment 13: It will be nice to include information on the study response rate.

• Response to reviewer: Unfortunately, this data is not available.

• Comment 14: A copy of the study questionnaire should be provided as supplementary material.

• Response to reviewer: We provided the study questionnaire as supplementary material (S1 Appendix). We categorized the answers into different sections during the analysis process to clarify the results.

• Comment 15: The layout of the methods needs improvement to ensure clarity. For example, the quantitative and qualitative components of the study can be described side by side in a logical and clear fashion. This will help avoid the creation of two subsections on data-analysis

• Response to reviewer: Thank you for your suggestion. We combined the subheadings accordingly and explained both components side by side when appropriate. For clarity, we split the heading of the data collection to help readers understand the data collection process in a more simplified way.

Results:

• Comment 16: Revisions to correct typos and all grammatical errors

• Response to reviewer: Thank you for your comment. We had the manuscript reviewed for appropriate language use and to correct all grammatical errors.

• Comment 17: The sentence: “Three distinct but ……………….” should have “and’’ connecting the third reasons listed.

• Response to reviewer: We modified this sentence.

• Comment 18: The authors are encouraged to revise the presentation of the data collected from the qualitative assessment. The results of the qualitative data are currently difficult to follow and to lengthy

• Response to reviewer: Thank you for these valuable suggestions. We revised the results as suggested to clarify the data and minimize the word count.

• Comment 19: Table legend should be provided to describe all abbreviations used in the table

• Response to reviewer: Thank you for pointing this out. We described and removed all abbreviations in the table.

Discussion:

• Comment 20: The use of the phrase, “catastrophic health expenditures” sounds somewhat awkward.

• Response to reviewer: Thank you for your feedback. There are a lot of research studies conducted using this equation and it is discussed on the WHO website. We believe replacing the phrase CHE can change the interpretation of the study. We used the definition of CHE in the discussion.

• Comment 21: In the absence of a clearer presentation of the shortcomings associated with receiving care from health centers that are closer to the residence of the study participants residence, the caregiver’s preference in attending these centers for treatment compared to the tertiary center seems reasonable.

• Response to reviewer: We did not address the shortcomings associated with each health facility in our quantitative study. This will be included in our limitations, as we believe that this topic can be explored in-depth in the future.

• Comment 22: The cross-sectional study design, a lack of hypothesis-driven sample size estimations or power calculation, and data was collected in one tertiary hospital limit the generalization of the study findings. This should be included the study shortcomings.

• Response to reviewer: We included this discussion in the limitations paragraph and elaborated further on this study’s limitations.

In addition to the above comments, all spelling and grammatical errors pointed out by the reviewers have been corrected. We look forward to hearing from you in due time regarding our submission and to respond to any further questions and comments you may have.

Kind regards

---

## [Decision Letter · Decision Letter 1]

24 Feb 2022

PONE-D-20-34450R1

Barriers to Seeking Care for Adolescents with Sickle Cell Disease in a Tertiary Care Hospital in Bahrain: Caregiver Experiences

PLOS ONE

Dear Dr. %Ahmed%,

Thank you for submitting your manuscript to PLOS ONE. After careful consideration, we feel that it has merit but does not fully meet PLOS ONE’s publication criteria as it currently stands. Therefore, we invite you to submit a revised version of the manuscript that addresses the points raised during the review process.

Please respond to the comments made by second reviewer

We look forward to receiving your revised manuscript.

Kind regards,

Mary Hamer Hodges, MBBS MRCP DSc

Academic Editor

PLOS ONE

Journal Requirements:

Reviewers' comments:

Reviewer's Responses to Questions

**Comments to the Author**

1. If the authors have adequately addressed your comments raised in a previous round of review and you feel that this manuscript is now acceptable for publication, you may indicate that here to bypass the “Comments to the Author” section, enter your conflict of interest statement in the “Confidential to Editor” section, and submit your "Accept" recommendation.

Reviewer #2: All comments have been addressed

Reviewer #3: (No Response)

2. Is the manuscript technically sound, and do the data support the conclusions?

Reviewer #2: Yes

Reviewer #3: Yes

3. Has the statistical analysis been performed appropriately and rigorously? 

Reviewer #2: Yes

Reviewer #3: Yes

4. Have the authors made all data underlying the findings in their manuscript fully available?

Reviewer #2: Yes

Reviewer #3: Yes

5. Is the manuscript presented in an intelligible fashion and written in standard English?

Reviewer #2: Yes

Reviewer #3: Yes

6. Review Comments to the Author

Reviewer #2: This is to thank the authors for the revisions conducted.

I wonder if a reference (a primary citation if available) can be provided for the phrase, “catastrophic health expenditures” where it was fist used in the main text to help readers understand that it is a common construct.

Reviewer #3: This study aims to evaluate the caregivers experiences of caring for an adolescent child with SCD in terms of access to health care, as well as the impact on the caregiver’s own emotional, psychological, financial and physical well-being.

The study is clear, with conclusions supported by data generated through this research. There could be more emphasis on implications of the findings on policy change (i.e. in the hospital) or suggested improvements in terms of support that can be provided for caregivers since the negative impact on caregivers’ psychosocial and physical wellbeing is evident in the findings. Some suggestions are highlighted briefly in the conclusion section but this could be expanded on. For example, is there scientific evidence that suggests that formal support options are beneficial to caregivers? Such as support groups, other forms of psychosocial support/counseling? There are a few typographical errors that need to be addressed.

Please take note of the comments and questions below for minor revisions:

1. Authors are encouraged to review referencing of statements:

a. Line 36-38 – Can you reference the statistic of SCD prevalence globally?

b. Line 42-44 – Can you reference the statistic concerning contribution of SCD to hospital admissions in Bahrain?

c. Line 58 – Can you reference the studies that note that caregivers experience shock?

2. Authors are encouraged to revise text that duplicates, is unclear or contradictory:

a. Compare lines 50-52 with lines 53-57. The former is captured in the latter. The latter would be sufficient. The authors are encouraged to revise these lines.

b. Line 59 – States that premarital testing for SCD is non-mandatory. Is this contradicted in lines 124-125? The authors are encouraged to review these statements and revise as needed.

c. Line 89-90 – The study is about the caregivers’ experiences, but this statement refers to the patients only. Either omit, or include caregivers. Please review.

d. Line 147 – Please clarify the years stated on this line: ‘in 2006 – 2001’ – should this read 2001 – 2006?

e. Line 177 – Please review. Suggest to change ‘with’ to ‘by’ or ‘conducted by’ to make it clear that it was the students collecting the data

f. Line 236 – Please review this line as it lacks clarity. …‘interviewers the researchers…’ – is there a comma missing or should the word interviewers be omitted?

g. Line 289 – change ‘w’ to ‘with’

h. Line 300-301 – ‘and 4% of caregivers received any other form of support from their families’ – lacks clarity – what is ‘any other form of support’

i. Line 301-302 – ‘Most caregivers (73%) had a low impact on their relationships with others.’ Incomplete statement – what had a low impact? Should this read ‘For most caregivers (73%), caring for an adolescent with SCD had a low impact on their relationships with others’?

j. Line 304 = ‘their patients’ – suggest to change this to ‘their children with SCD’ or ‘their affected children’

3. Authors are encouraged to include definitions to terms such as adolescents and catastrophic health expenditure, with appropriate referencing:

a. Line 70 – The authors are encouraged to define ‘Catastrophic health expenditures’ in this section. If using an official WHO definition, this should include a reference.

b. Line 96 – The study refers to adolescents. Authors are encouraged to review the need to include a definition of the adolescent age group in this section, providing a reference.

4. The authors are encouraged to review the final statements of the introductory section. Is there a broader goal/impact that this study might have, which can be stated here? For example, to provide additional support to caregivers, improve access to care etc. What do the authors plan to address or achieve through this study? This can then be expanded on in the conclusion section based on findings from the study.

5. Selection of participants – The authors are encouraged to review this section to address how the participants for the qualitative study were selected? This is referred to later under ‘Study Design and Sample’ but should be included in the ‘selection of participants’ section.

7. PLOS authors have the option to publish the peer review history of their article (what does this mean?). If published, this will include your full peer review and any attached files.

Reviewer #2: **Yes: **Andrew T. Olagunju

Reviewer #3: No

---

## [Author Response · Author response to Decision Letter 1]

6 Mar 2022

Reference number: PONE-D-20-34450

Dear Editor,

Thank you for allowing us to re-submit a revised draft of our manuscript titled [Barriers to Seeking Care for Adolescents with Sickle Cell Disease in a Tertiary Care Hospital in Bahrain: Caregiver Experiences] to PLOS ONE. We appreciate the time and effort that you and the reviewers have dedicated to providing your valuable feedback on our manuscript. We are grateful to the reviewers for their generous comments on our paper. We have been able to incorporate changes to reflect most of the suggestions provided by the respectful reviewers and revised the manuscript accordingly. We have highlighted these changes within the manuscript. Our responses to these comments are given under each of your comment in italics.

Journal Requirements:

Response to Editor: Thank you for your concern. We have ensured that all our cited papers are not retracted. We have added new references in our revised manuscript as required by the reviewers.

Reviewers' comments:

Reviewer's Responses to Questions

Comments to the Author

1. If the authors have adequately addressed your comments raised in a previous round of review and you feel that this manuscript is now acceptable for publication, you may indicate that here to bypass the “Comments to the Author” section, enter your conflict of interest statement in the “Confidential to Editor” section, and submit your "Accept" recommendation.

Reviewer #2: All comments have been addressed Reviewer #3: (No Response)

2. Is the manuscript technically sound, and do the data support the conclusions?

Reviewer #2: Yes

Reviewer #3: Yes

3. Has the statistical analysis been performed appropriately and rigorously?

Reviewer #2: Yes

Reviewer #3: Yes

4. Have the authors made all data underlying the findings in their manuscript fully available?

Reviewer #2: Yes

Reviewer #3: Yes

5. Is the manuscript presented in an intelligible fashion and written in standard English?

Reviewer #2: Yes

Reviewer #3: Yes

6. Review Comments to the Author

Reviewer #2: This is to thank the authors for the revisions conducted.

I wonder if a reference (a primary citation if available) can be provided for the phrase, “catastrophic health expenditures” where it was fist used in the main text to help readers understand that it is a common construct.

Response to Reviewer: Thank you for this suggestion. We have provided references of catastrophic health expenditures definition in the introduction.

Reviewer #3: This study aims to evaluate the caregivers experiences of caring for an adolescent child with SCD in terms of access to health care, as well as the impact on the caregiver’s own emotional, psychological, financial and physical well-being.

The study is clear, with conclusions supported by data generated through this research. There could be more emphasis on implications of the findings on policy change (i.e. in the hospital) or suggested improvements in terms of support that can be provided for caregivers since the negative impact on caregivers’ psychosocial and physical wellbeing is evident in the findings. Some suggestions are highlighted briefly in the conclusion section but this could be expanded on. For example, is there scientific evidence that suggests that formal support options are beneficial to caregivers? Such as support groups, other forms of psychosocial support/counseling? There are a few typographical errors that need to be addressed.

Response to Reviewer: Thank you for your suggestion. We have expanded the suggestions in the conclusion section. We have provided as well scientific evidence that suggest formal support can be of beneficial to caregivers. We have addressed typographical errors in the conclusion section.

Please take note of the comments and questions below for minor revisions:

1. Authors are encouraged to review referencing of statements:

a. Line 36-38 – Can you reference the statistic of SCD prevalence globally?

Response to Reviewer: Thank you for this suggestion. Quoted from the first study referenced (keto et al) “The number of all-age individuals affected by SCA globally is currently unknown and cannot be estimated reliably owing to the paucity of epidemiological data, in particular mortality data, in areas of high prevalence.” 

b. Line 42-44 – Can you reference the statistic concerning contribution of SCD to hospital admissions in Bahrain?

Response to Reviewer: Thank you for this suggestion. We have added the references concerning the contribution of SCD to hospital admissions in Bahrain.

c. Line 58 – Can you reference the studies that note that caregivers experience shock?

Response to Reviewer: Thank you for this suggestion. It was only one study in our references. 

2. Authors are encouraged to revise text that duplicates, is unclear or contradictory:

a. Compare lines 50-52 with lines 53-57. The former is captured in the latter. The latter would be sufficient. The authors are encouraged to revise these lines.

Response to Reviewer: Thank you for this suggestion. We have removed this sentence as the latter is sufficient.

b. Line 59 – States that premarital testing for SCD is non-mandatory. Is this contradicted in lines 124-125? The authors are encouraged to review these statements and revise as needed.

Response to Reviewer: In 2004, premarital SCD testing became mandatory in Bahrain. However, the results of the premarital test does not their decision to marry by law. A new citation was added in the methods sections. Line 59 involves another country. We have removed this sentence.

c. Line 89-90 – The study is about the caregivers’ experiences, but this statement refers to the patients only. Either omit, or include caregivers. Please review. 

Response to Reviewer: Thank you for this suggestion. We have removed this sentence.

d. Line 147 – Please clarify the years stated on this line: ‘in 2006 – 2001’ – should this read 2001 – 2006?

Response to Reviewer: Thank you for this suggestion. We have clarified the years to read like this 2001-2006.

e. Line 177 – Please review. Suggest to change ‘with’ to ‘by’ or ‘conducted by’ to make it clear that it was the students collecting the data

Response to Reviewer: Thank you for this suggestion. We have made the sentence clear as the students were data collectors.

f. Line 236 – Please review this line as it lacks clarity. …‘interviewers the researchers…’ – is there a comma missing or should the word interviewers be omitted?

Response to Reviewer: Thank you for this suggestion. We have removed the word interviewers.

g. Line 289 – change ‘w’ to ‘with’

Response to Reviewer: Thank you for this suggestion. We have changed this word.

h. Line 300-301 – ‘and 4% of caregivers received any other form of support from their families’ – lacks clarity – what is ‘any other form of support’

Response to Reviewer: Thank you for your suggestion. We have clarified that sentence as most caregivers mentioned accompanying their affected child as other form of support from their families.

i. Line 301-302 – ‘Most caregivers (73%) had a low impact on their relationships with others.’ Incomplete statement – what had a low impact? Should this read ‘For most caregivers (73%), caring for an adolescent with SCD had a low impact on their relationships with others’?

Response to Reviewer: Thank you for your suggestion. Yes. We have clarified that sentence.

j. Line 304 = ‘their patients’ – suggest to change this to ‘their children with SCD’ or ‘their affected children’

Response to Reviewer: Thank you for your suggestion. We have changed this to “their affected children 

3. Authors are encouraged to include definitions to terms such as adolescents and catastrophic health expenditure, with appropriate referencing:

a. Line 70 – The authors are encouraged to define ‘Catastrophic health expenditures’ in this section. If using an official WHO definition, this should include a reference.

Response to Reviewer: Thank you for this suggestion. We have provided references of catastrophic health expenditures definition in the introduction.

b. Line 96 – The study refers to adolescents. Authors are encouraged to review the need to include a definition of the adolescent age group in this section, providing a reference.

Response to Reviewer: Thank you for this suggestion. We have added the definition of adolescent age group according to WHO with a reference from the official Brtannica web page.

The authors are encouraged to review the final statements of the introductory section. Is there a broader goal/impact that this study might have, which can be stated here? For example, to provide additional support to caregivers, improve access to care etc. What do the authors plan to address or achieve through this study? This can then be expanded on in the conclusion section based on findings from the study.

Response to Reviewer: Thank you for this suggestion. We have added the impact that the authors aimed to achieve in the final introduction statement. 

Selection of participants – The authors are encouraged to review this section to address how the participants for the qualitative study were selected? This is referred to later under ‘Study Design and Sample’ but should be included in the ‘selection of participants’ section.

Response to Reviewer: Thank you for your inquiry. We have moved the additional sampling technique from study design and sample section to the selection of participants’ section. The selection of qualitative and quantitative were both consecutive sampling except some quantitative participants were added by snowball sampling as mentioned in the selection of participant section.

---

## [Decision Letter · Decision Letter 2]

23 Mar 2022

Caregivers’ experience of seeking Care for Adolescents with Sickle Cell Disease in a Tertiary Care Hospital in Bahrain

PONE-D-20-34450R2

Dear Dr. %Ahmed%,

We’re pleased to inform you that your manuscript has been judged scientifically suitable for publication and will be formally accepted for publication once it meets all outstanding technical requirements.

Kind regards,

Mary Hamer Hodges, MBBS MRCP DSc

Academic Editor

PLOS ONE

Additional Editor Comments (optional):

Thank you for this manuscript on this important topic and for your revisions.

Reviewers' comments:

Reviewer's Responses to Questions

**Comments to the Author**

1. If the authors have adequately addressed your comments raised in a previous round of review and you feel that this manuscript is now acceptable for publication, you may indicate that here to bypass the “Comments to the Author” section, enter your conflict of interest statement in the “Confidential to Editor” section, and submit your "Accept" recommendation.

Reviewer #3: All comments have been addressed

2. Is the manuscript technically sound, and do the data support the conclusions?

Reviewer #3: Yes

3. Has the statistical analysis been performed appropriately and rigorously? 

Reviewer #3: Yes

4. Have the authors made all data underlying the findings in their manuscript fully available?

Reviewer #3: Yes

5. Is the manuscript presented in an intelligible fashion and written in standard English?

Reviewer #3: Yes

6. Review Comments to the Author

Reviewer #3: The comments from the prevision review have been addressed sufficiently.

I don't have any additional comments at this time.

7. PLOS authors have the option to publish the peer review history of their article (what does this mean?). If published, this will include your full peer review and any attached files.

Reviewer #3: No

---

## [Editor Report · Acceptance letter]

30 Mar 2022

PONE-D-20-34450R2 

Caregivers’ experience of seeking Care for Adolescents with Sickle Cell Disease in a Tertiary Care Hospital in Bahrain 

Dear Dr. Ahmed:

I'm pleased to inform you that your manuscript has been deemed suitable for publication in PLOS ONE. Congratulations! Your manuscript is now with our production department. 

Kind regards, 

on behalf of

Prof. Mary Hamer Hodges 

Academic Editor

PLOS ONE